# Deciphering the mechanism of anhydrobiosis in the entomopathogenic nematode *Heterorhabditis indica* through comparative transcriptomics

**Manimaran Balakumaran** [1]☯*, **Parameshwaran Chidambaranathan**[2], **Jagannadham Prasanth Tej Kumar J. P.**[2], **Anil Sirohi**[1]*, **Pradeep Kumar Jain**[2], **Kishore Gaikwad**[2], **Yuvaraj Iyyappan**[2], **Atmakuri Ramakrishna Rao**[3], **Sarika Sahu**[3], **Anil Dahuja**[4], **Sharad Mohan** [1]☯*

1 Division of Nematology, ICAR-Indian Agricultural Research Institute, New Delhi, India, 2 ICAR-National Institute for Plant Biotechnology, New Delhi, India, 3 Centre for Agricultural Bioinformatics, ICAR-Indian Agricultural Statistics Research Institute, New Delhi, India, 4 Division of Biochemistry, ICAR-Indian Agricultural Research Institute, New Delhi, India

☯ These authors contributed equally to this work.
* marannkl@gmail.com (MB); sharad@iari.res.in (SM); anilsirohi@yahoo.com (AS)

**Data Availability Statement:** All available data related to the manuscript are attached as supporting information.

## Abstract

The entomopathogenic nematode, *Heterorhabditis indica*, is a popular biocontrol agent of high commercial significance. It possesses tremendous genetic architecture to survive desiccation stress by undergoing anhydrobiosis to increase its lifespan—an attribute exploited in the formulation technology. The comparative transcriptome of unstressed and anhydrobiotic *H. indica* revealed several previously concealed metabolic events crucial for adapting towards the moisture stress. During the induction of anhydrobiosis in the infective juveniles (IJ), 1584 transcripts were upregulated and 340 downregulated. As a strategy towards anhydrobiotic survival, the IJ showed activation of several genes critical to antioxidant defense, detoxification pathways, signal transduction, unfolded protein response and molecular chaperones and ubiquitin-proteasome system. Differential expression of several genes involved in gluconeogenesis - β-*oxidation of fatty acids*, *glyoxylate pathway*; glyceroneogenesis; fatty acid biosynthesis; amino-acid metabolism - *shikimate pathway*, *sachharopine pathway*, *kyneurine pathway*, *lysine biosynthesi*s; one-carbon metabolism—*polyamine pathway*, *transsulfuration pathway*, *folate cycle*, *methionine cycle*, *nucleotide biosynthesis*; mevalonate pathway; and glyceraldehyde-3-phosphate dehydrogenase were also observed. We report the role of shikimate pathway, sachharopine pathway and glyceroneogenesis in anhydrobiotes, and seven classes of repeat proteins, specifically in *H. indica* for the first time. These results provide insights into anhydrobiotic survival strategies which can be utilized to strengthen the development of novel formulations with enhanced and sustained shelf-life.

**Funding:** "the first author Manimaran, B. carried out this work as part of his PhD research partly funded by the Post Graduate School, ICAR-IARI fellowship (Award Number 10156) The fellowship largely supported the lodging and subsistence, and a small part covered the basic contingencies for the research. The funders had no role in study design, data collection and analysis, decision to publish, or preparation of the manuscript. There was no additional external funding received for this study."

**Competing interests:** The authors have declared that no competing interests exist.

## 1. Introduction

Nematode is a ubiquitous metazoan group which has evolved adaptations to survive the environmental extremities in which they habituate. Soil environment especially agricultural fields are exposed to different levels of moisture availability. The 'anhydrobiosis' term first proposed by [1], is a strategy of an organism to survive extreme desiccation. It is a 'state of suspended animation' with little to undetectable metabolism [2, 3], governed by complex molecular mechanisms interlinking various physiological and biochemical cascade to keep the organism alive. Anhydrobiosis was first described by [4] from bdelloid rotifer *Philodina roseola* and later described in different taxonomic categories ranging from yeast to higher plants; from rotifers to sleeping chironomid [5–7]. Depending on the degree of moisture stress several nematodes are also reported to possess varied level of anhydrobiotic potential to survive. Entomopathogenic nematodes (EPN) colonize in soil and have tremendous ability to kill a wide range of insect pests and regarded as an alternative for chemical pesticides [8, 9]. The third-stage juvenile or infective juveniles (IJ) of the EPN is the only stage that survives outside the host often exposed to desiccation stress. They possess the ability to withstand desiccation by entering into anhydrobiosis [10, 11] an attribute which has been successfully exploited for the development of stable commercial formulations of EPN.

Numerous attempts have been made to decipher the mechanisms involved in anhydrobiosis in various organisms including nematodes [12–21]. They adopt two set of strategies—one is to prevent the water loss and the other is to repair the damage caused. Nematodes follow various behavioral responses like coiling and clumping [22, 23]; physiological adaptation involving the replacement of bound water and utilization of alternate pathways, like glyoxylate pathway, to satisfy energy requirements during entry and exit of anhydrobiosis [12, 24]; and biochemical adaptations like accumulation of sugars (trehalose), synthesis of heat-shock proteins (Hsps), hydrophilic proteins and modification of fatty acid metabolism [13, 18, 25]. The success of survival not only depends on the capacity to enter and exit the desiccation, but also in the potential to repair the damages caused during dehydration and rehydration.

*Heterorhabditis indica* (Nematoda; Family: Heterorhabditidae) is an EPN native to the Indian sub-continent having high commercial significance in the International biopesticide market. Understanding the moisture stress tolerance is essential not only to improve the quality and shelf-life of the commercially available dry-formulations, but also to strengthen the knowledge-base which can be applicable in related research fields. Though there are many recent reports in other organisms with the advancements in sequencing technologies, comprehensive information on anhydrobiotic survival strategies adopted by EPN is lacking. In the present study we used comparative transcriptomics between moisture stressed and unstressed *H. indica* IJ to decipher the enigma of anhydrobiotic survival mechanism. We highlight the uniqueness of certain metabolic pathways that are altered during anhydrobiosis.

## 2. Results

### 2.1 Transcriptome general description

A comparative transcriptome sequencing and assembly of unstressed and anhydrobiotic IJ of *H. indica* resulted in 60,589,834 and 51,151,316 paired-end reads, respectively. The GC content in the unstressed IJ was 41.7% compared to 40.1% in anhydrobiotic, while the read lengths in both the samples averaged at 81bp (S1A and S1B Table). The trimmed reads further assembled using *Trinity* yielded 93,932 transcripts from both the samples. The Phred score was above Q30 (error-probability $\geq$ 0.001) for ~85% of bases. All assembled transcripts were >200bp. Maximum of 13.56% assembled transcripts were between 2000–3000 bp, followed by 11.16% and 11.12% between 1000–1500 bp and 3000–5000 bp, respectively (Fig 1). The short

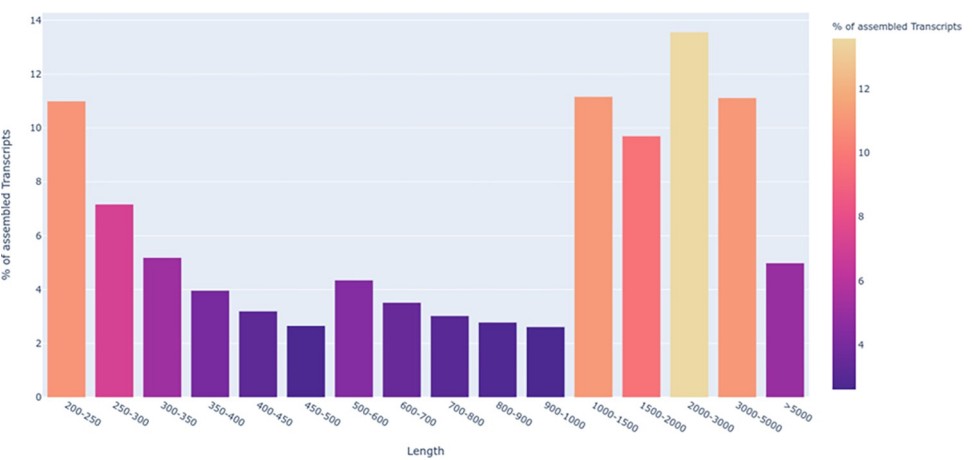

**Fig 1. Assembled transcript length distribution of *H. indica* transcriptome.**

transcripts ranged between 200 and 500 bp corresponding to 10.99% transcript assemblies and longest size of >5000 bp were assembled for 4.98% of the transcripts. Further, we focused on ≥200 bp for transcript expression estimation and downstream annotations (S2 Table).

## 2.2 Transcriptome quality analysis and completeness

The *TransRate* analysis metrics showed 93932 contigs and the transcript size ranged between 201 and 22658 nucleotides. The N50 value of the transcriptome was 2843 and no single contigs shorter than 200 nucleotides (n_under_200 = 0) or ambiguous bases (N) were identified. CEGMA (Core Eukaryotic Genes Mapping Approach) analysis yielded 230 complete and 18 missing transcripts out of 248 core eukaryotic genes indicating 92.74% recovery in the transcriptome assembly, thus, confirming the high quality of the transcriptome. Eukaryota and Nematoda were used as lineages for BUSCO (Benchmarking Universal Single-Copy Orthologs) analysis which resulted in 79.2% complete BUSCOs, while the fragmented and missing BUSCOs were 12.2% and 8.6%, respectively, for the eukaryote lineage. For the nematode lineage, 72.2% complete BUSCOs were obtained, while fragmented and missing BUSCOs were 5.7% and 22.1%, respectively. The results of all three tools are given in (S3–S5B Tables).

Further, comparative analysis of our assembly with available omics data on *H. indica* revealed it to be relatively superior. The values for N50, N60, N70, N80, N90 and N100 were more than double; clean with no ambiguous bases; and without gaps as compared to 1670 ambiguous bases and 66 gaps reported by Somvanshi *et al.* (2016) [26] (S6 Table)

We aligned the raw data obtained for fresh and anhydrobiotic nematodes with the draft genome of Bhat *et al.* (2022) [27] as the reference genome. Alignment reads using *HISAT2* was 73.79% for the former and 90.29% for the latter as compared to 0.02% with Somvanshi *et al.*, (2016) [26]. Bowtie2 resulted in 58.12% for fresh and 67.34% for anhydrobiotic samples whereas only 0.05% for Somvanshi *et al.*, (2016) [26].

Reference-based mapping with Bhat *et al.* (2022) [27] revealed 78,454 transcripts mapped to it. Mapping the transcriptome of Somvanshi *et al.*, (2016) [26] over Bhat *et al.*, (2022) [27] resulted in less than 1% mapping.

## 2.3 Gene Expression Estimation

The summary of read alignment and expression estimation are given in (S7 Table). Read alignment using *Bowtie 2* showed about 86.0% of the filtered reads (paired-end) from both samples

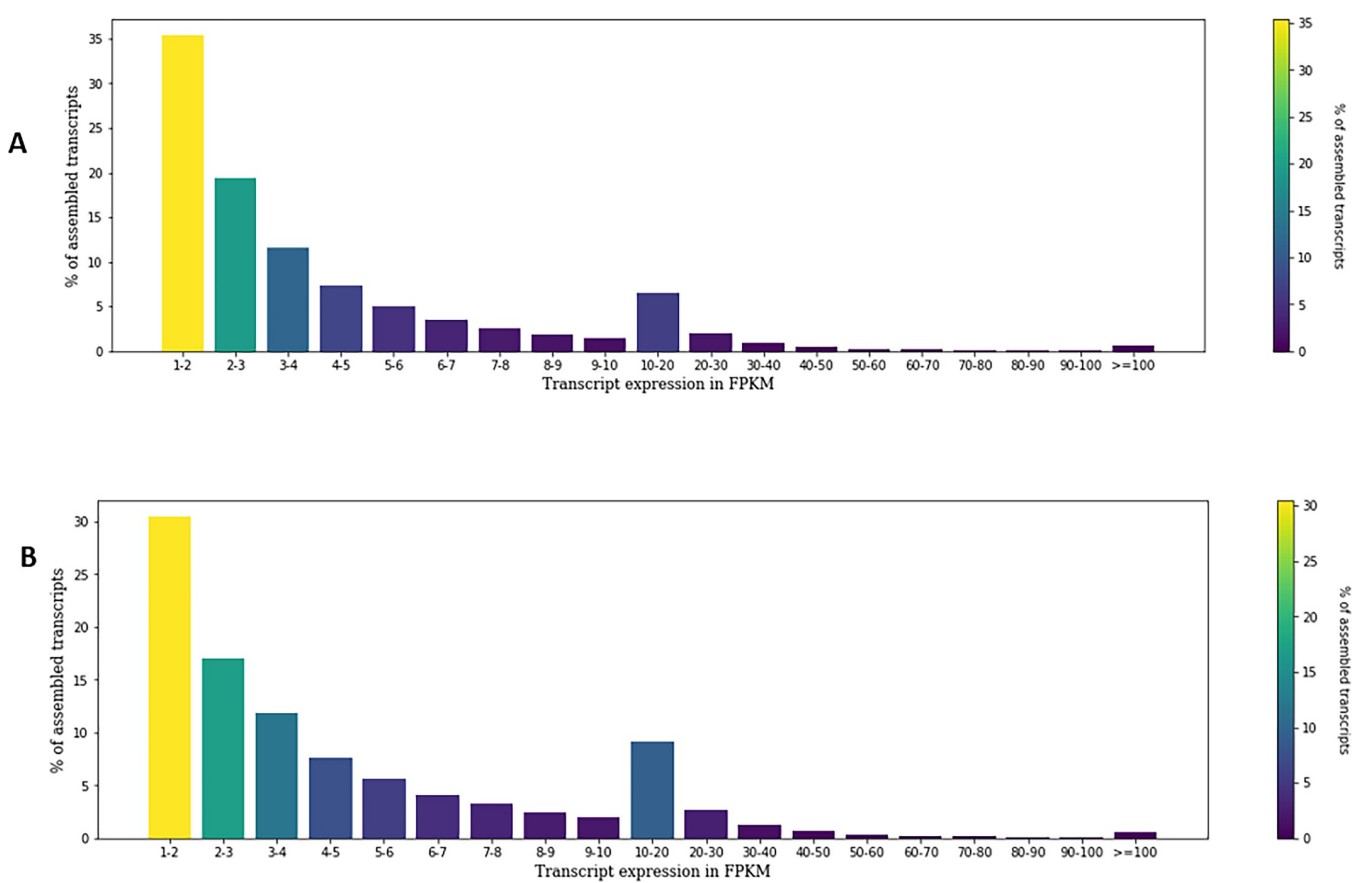

**Fig 2. Transcript expression distribution of (A) unstressed and (B) anhydrobiotic *H. indica* IJ.**

were aligned to the assembled transcriptome, wherein unstressed IJs provided 88.82% alignment as compared to 84.23% in anhydrobiotic IJ. Further, based on the Fragments Per Kilo Million bases (FPKM) value, 52,678 unique transcripts out of 93932 had >1 FPKM, and these transcripts were used for further downstream analysis. Individually 39996 transcripts from unstressed IJ and 48007 transcripts from anhydrobiotic IJs had >1 FPKM. Out of which 38.3% and 36.6% ranged between 2 and 5 FPKM, followed by 35.3% and 30.4% between 1–2 FPKM in unstressed and anhydrobiotic IJ, respectively (Fig 2; S8 Table).

## 2.4 Characterization of *H. indica* transcripts

The assembled *H. indica* transcripts were compared with NCBI non-redundant protein database using BLASTX program. Matches with E-value $\leq 10^{-5}$ and similarity score $\geq 40\%$ were retained for further annotation. We found ~ 44,787 (85.02%) of assembled transcripts had at least one significant hit in NCBI database. A maximum of 36,880 gene ontology terms were identified for molecular functions, 18,378 for cellular components and 17,584 for biological functions. Around 74% had confidence level of at least $1e^{-5}$, while ~68% had similarity of $> 60\%$ at protein level with the available proteins in the NCBI database. The top hit matched with an animal-parasitic nematode, *Ancyclostoma ceylanicum* (Phylum: Nematoda, Order: Strongylida). The top 10 hits included four species of *Caenorhabditis* genus and *Pritionchus pacificus* (Order: Rhabditida), close associates of *H. indica* which also belongs to the same Order (Fig 3)

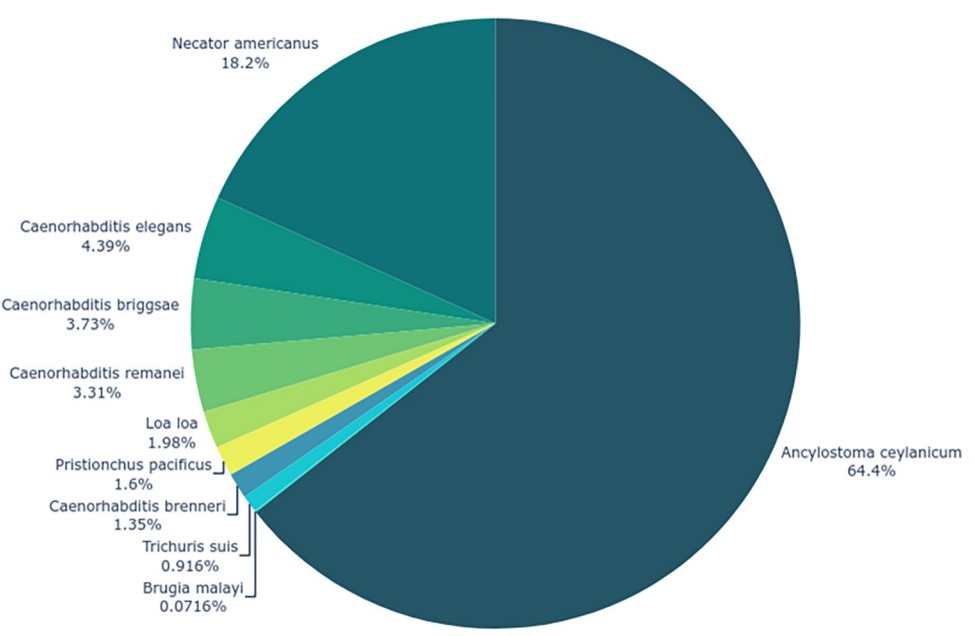

**Fig 3. Top ten BLASTX hits of *H. indica* with other nematodes.**

## 2.5 Functional annotation

Among the top 15 gene ontology (GO) terms with respect to biological processes, the translation [GO:0006412], DNA templated transcription [GO:0006351], and transmembrane transport [GO:0055085] occupied the top 3 positions with 604 (3.43%), 427 (2.42%) and 407 (2.31%) transcripts, respectively, while signal transduction including small GTPase mediated [GO:0007264] and transport [GO:0055085, GO:0016192, GO:0006886, GO:0015031, GO:0006810] were predominant. In molecular functions, ATP binding [GO:0005524] (2006), nucleic acid binding [GO:0003676] (1306) and metal ion binding [GO:0046872] (1208) were the top hits with 5.43%, 3.54% and 3.2% of transcripts, respectively. The genes for binding related activity [GO:0005524, GO:0003676, GO:0046872, GO:0008270, GO:0003700, GO:0003677, GO:0043565, GO:0000166] were most frequent, followed by protein kinase [GO:0004672], integral component of membrane [GO:0016021] and catalytic activity [GO:0003824]. The top 3 GO terms related to cellular components were represented by the genes involved with integral component of membrane [GO:0016021], nucleus [GO:0005634], and membrane [GO:0016020] with transcript numbers 2836 (15.43%), 958 (5.21%) and 485 (2.63%), respectively; while the G-protein coupled receptor activity [GO:0004930], ribosome [GO:0005840], intracellular [GO:0005622] and cytoplasm [GO:0005737] were the other most frequent cellular component GO terms (Fig 4, S1 File). Comparison between anhydrobiotic and unstressed IJ revealed ATP binding [GO:0005524] (193), integral component of membrane [GO:0016021] (101) and nucleus [GO:0005634] (88) as the top 3 upregulated GO terms, while structural component of ribosome [GO:0003735] (73), translation [GO:0006412] (70) and ribosome [GO:0005840] (61) were the top 3 downregulated gene ontologies (Fig 5, S2 File). Gene enrichment analysis of differentially expressed transcripts through Blast2GO revealed that 1202 GO terms were enriched in total with p-value and FDR value <0.05 (S9 Table, Fig 6.).

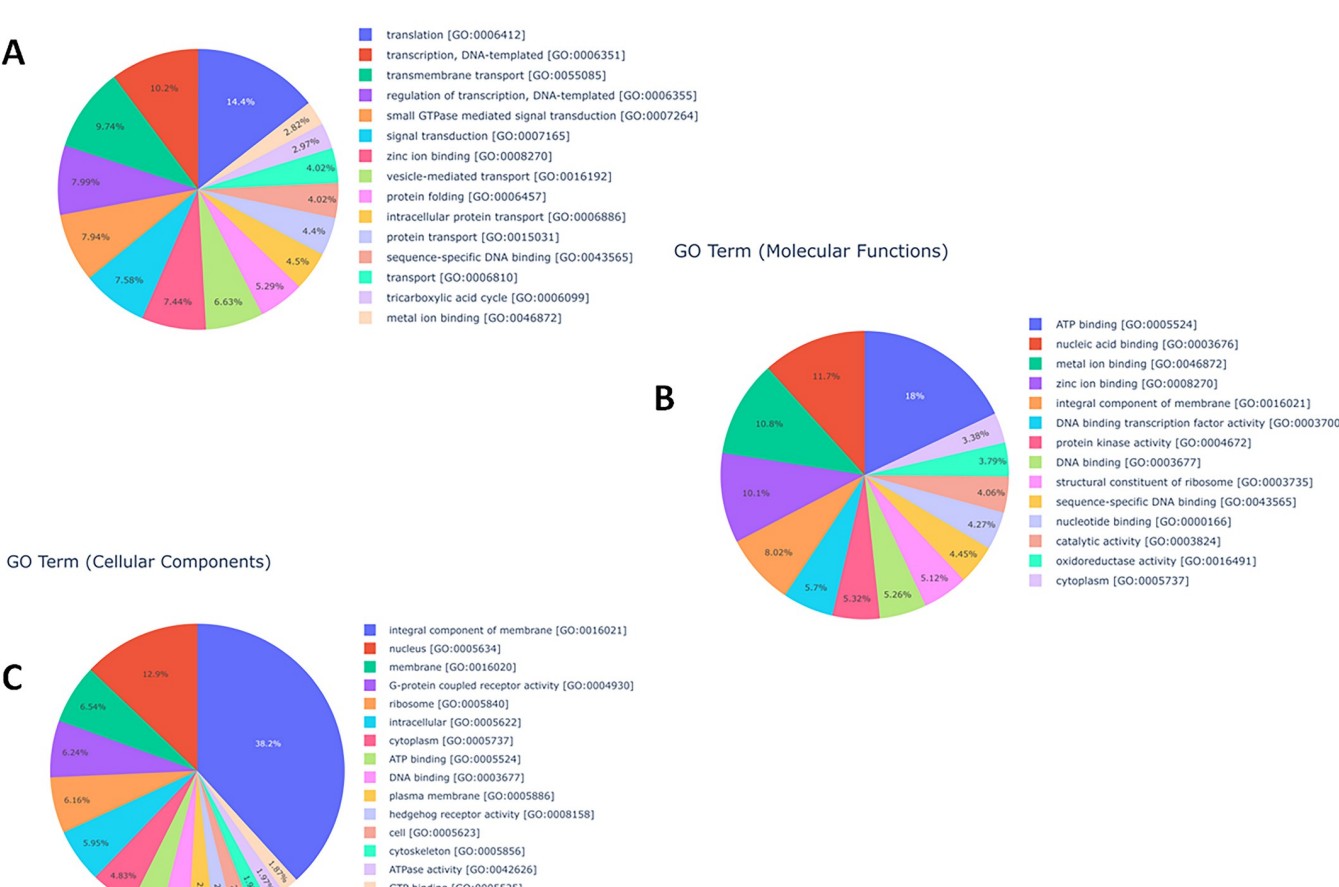

**Fig 4.** Top 15 gene ontology terms with respect to (A) biological processes (B) molecular functions (C) cellular components in the transcriptome of *H. indica*.

## 2.6 Differential expression analysis

Differential gene expression analysis was performed using DESeq program. Transcripts having read count ≥ 1 for both samples were chosen for differential expression analysis. Out of 93,933 1,924 (1584 upregulated and 340 downregulated) transcripts were differentially expressed (Fig 7).

## 2.7 Identifying desiccation stress response genes in anhydrobiotic *H. indica*

The moisture stress induced a cascade of 5 stress responsive pathways in anhydrobiotic IJ (Table 1, S3 File). These were represented by 15 upregulated transcripts involved in anti-oxidant defenses. We identified DNA damage-inducible protein-1 transcripts which did not significantly get upregulated in anhydobiotic IJ, although showed relatively higher FPKM value as compared to unstressed IJ (S1 Fig).

There were 27 upregulate and 6 downregulated transcripts within the detoxification pathways. In Phase I, 2 transcripts, each, of SDR were significantly upregulated and downregulated, while in Phase II, UGT was downregulated and GST was upregulated, with further upregulation observed in ATP-binding cassette (ABC class transporters) (S2 Fig). For signal transduction 94 upregulated and 10 downregulated transcripts were observed. There was a significant

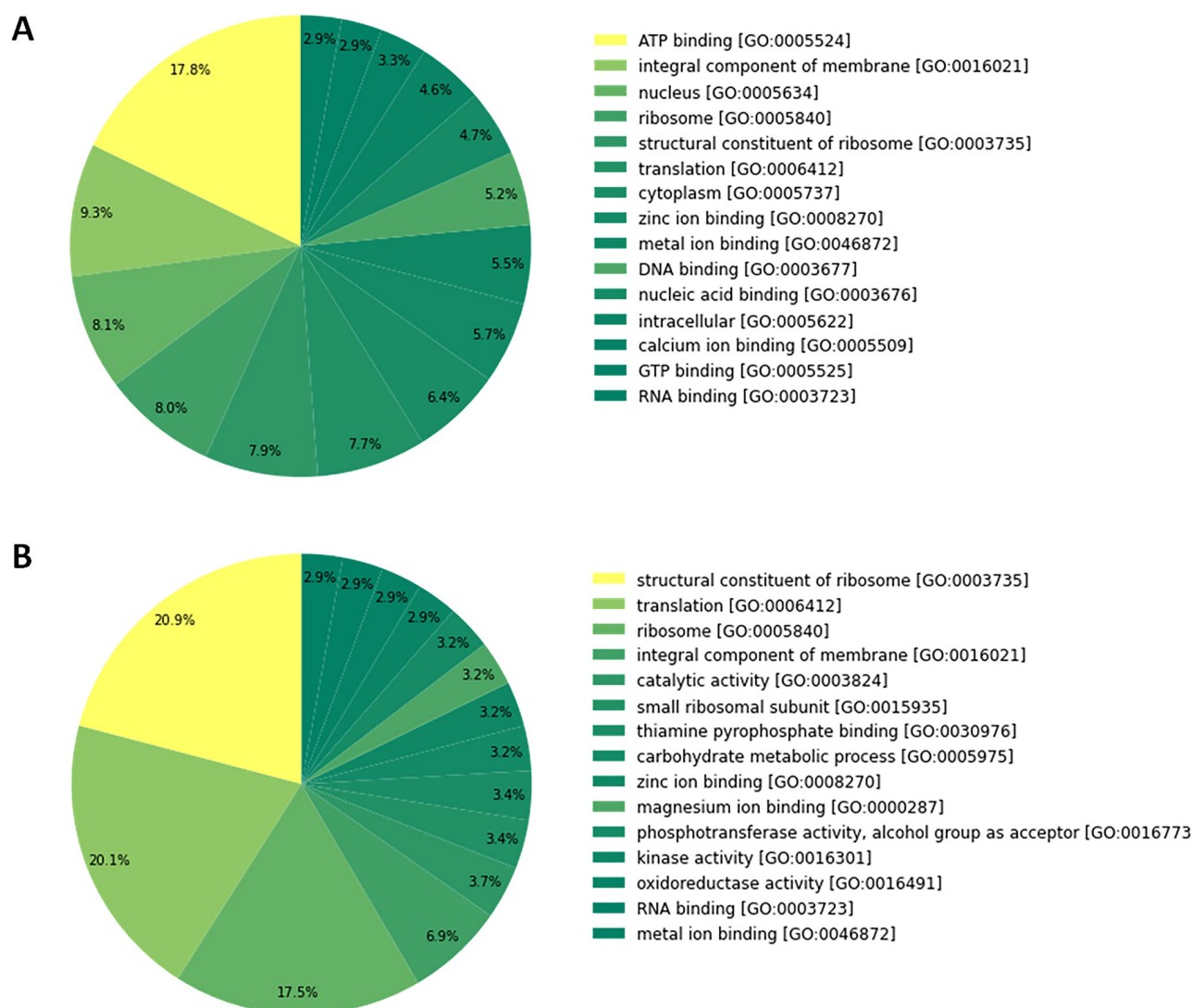

**Fig 5.** Top 15 gene ontology terms for transcripts differentially expressed in anhydrobiotic *H. indica* (A) Upregulated (B) Downregulated.

upregulation in phosphatidylinositol signal regulating genes wherein two transcripts each for *plc-1* gene, diacylglycerol kinase, protein kinase C and 1 for calcium ion binding protein encoding genes were upregulated. One transcript for mitogen-activated protein kinases (MAPK/p38), 41 upregulated small GTPases and 2 downregulated and 45 upregulated and 8 downregulated kinases and phosphatases were observed (S3 Fig).

For unfolded protein response and molecular chaperones, 34 upregulated transcripts were observed, predominated by TCP-1 chaperonin family (9), followed by protein glycosylation (7), Cyclophilin type of peptidyl-prolyl cis-trans isomerase (6), *pqn-95* and signal recognition particle (3 each). We identified significant upregulation in at least one transcript of *hsp60* and four of *hsp70* (A comprehensive discussion on the HSPs and sHSPs has been dealt in a separate manuscript). A single downregulation was observed for protein disulfide isomerase (S4 Fig).

Overall there were 32 upregulated transcripts observed in the ubiquitin-proteasome system. Significant upregulation in 4 transcripts involved in the ubiquitin tagging process namely— ubiquitin-activating enzyme (E1), ubiquitin-conjugating enzyme (E2) and ubiquitin ligases

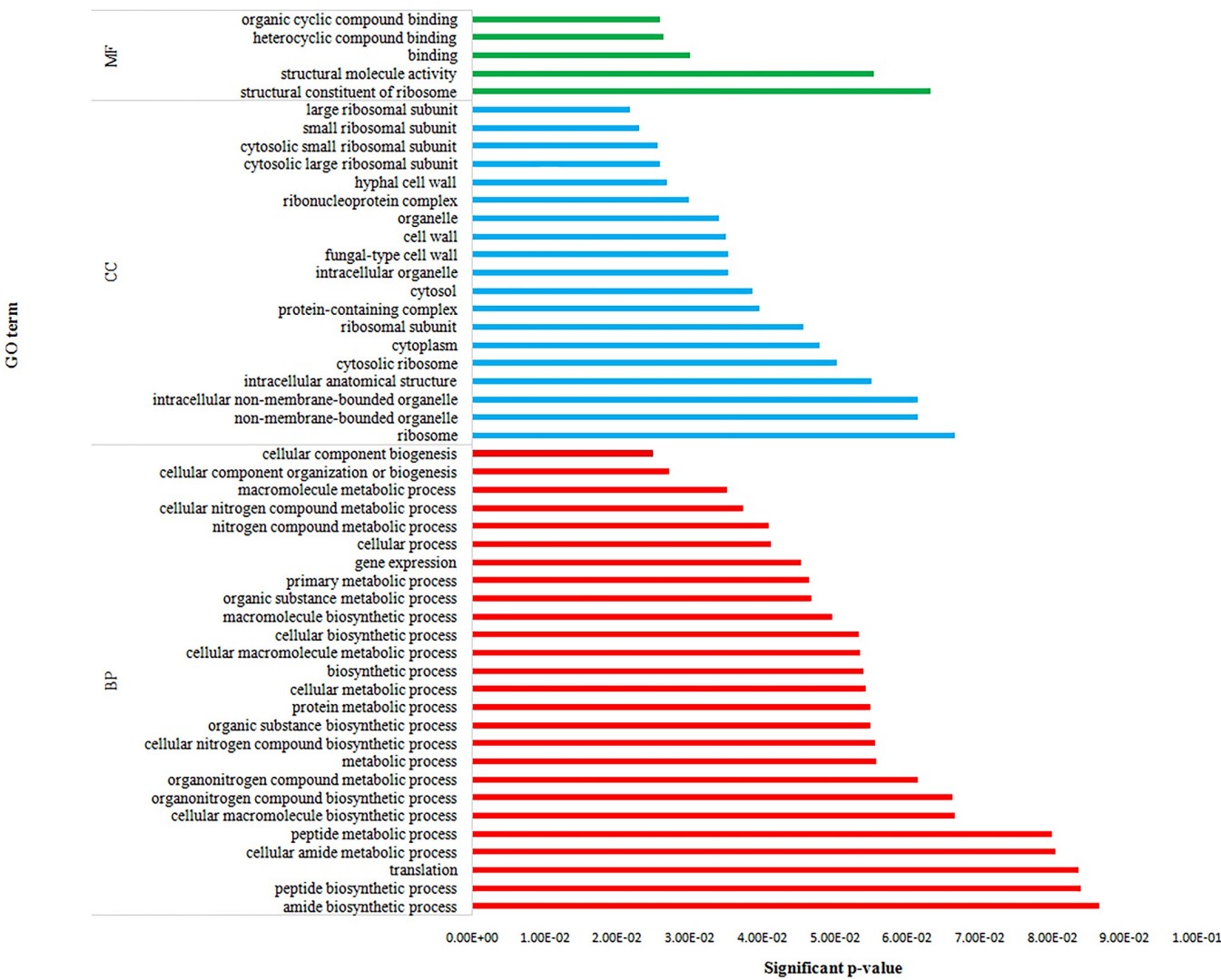

**Fig 6. Top 50 GO enriched terms for differentially expressed genes in anhydrobiotic *H. indica*.**

(E3); and 23 transcripts involved in 26S proteasome including the alpha-subunits like *pas-1*, *pas-2*, *pas-3*, *pas-4*, and beta- subunit *pbs-2* of 20S proteasome core and *Rpn1* and *Rpt5* of 19S proteasome regulatory subunits. Upregulated protein sorting transcripts comprised of AAA ATPase (4), *cdc-48.2* (1) and Cation transport ATPase (3) (S5 Fig).

## 2.8 Repeat proteins

Seven different classes of repeat proteins were upregulated. The predominant ones were HEAT class and Spectrin repeat-containing domain proteins (4 transcripts each), followed by WD-40 and Ankyrin class proteins (2 transcripts each) (Table 2, S6 Fig).

## 2.9 Identifying transcripts of biochemical pathways that are differentially expressed during desiccation stress

The moisture stress induced significant upregulations in several stress responsive biochemical pathways in anhydrobiotic IJ (Table 3, S4 File, S7–S13 Figs, and S1 Plate). These were

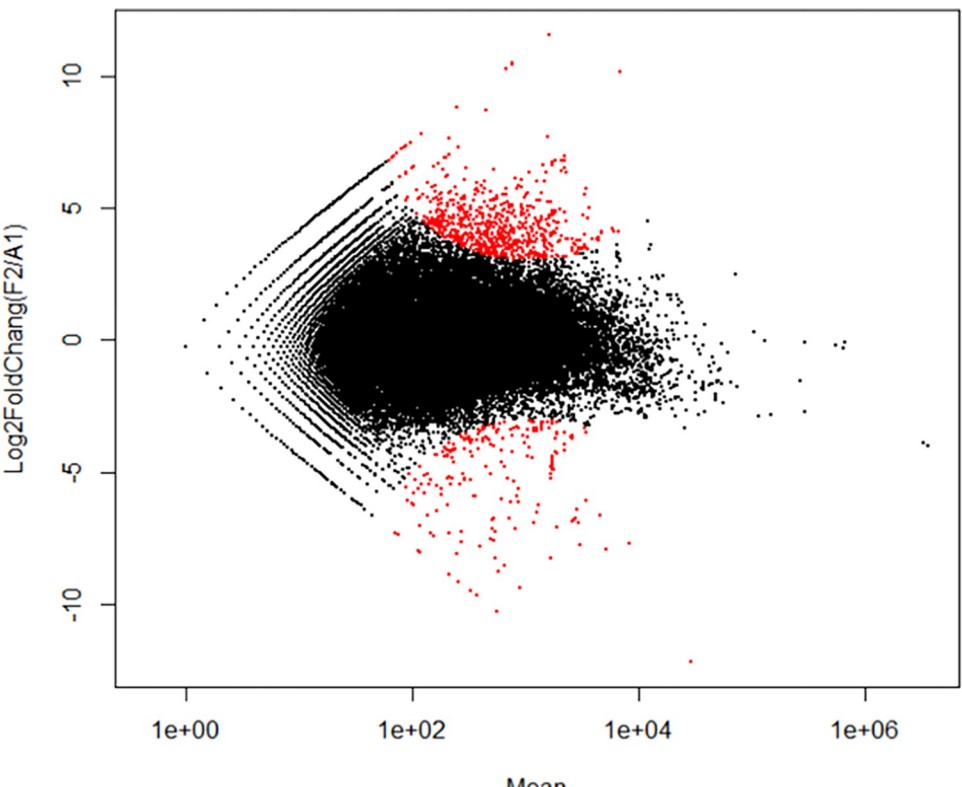

**Fig 7. Fold change and mean plot between unstressed and anhydrobiotic *H. indica* IJ using DESeq program (transcripts with p-value < 0.01 in red).**

represented by Fatty acid anabolism (9 upregulated, 1 downregulated); Gluconeogenesis (27 upregulated), primarily involving Beta-oxidation of fatty acids, Glyoxylate pathway and Glyceroneogenesis; Glyceraldehyde-3-phosphate dehydrogenase (1 upregulated, 4 downregulated); Mevalonate pathway (3 upregulated); Amino acid metabolism (9 upregulated) primarily involving Lysine biosynthesis, Saccharopine pathway, Shikimate pathway and Kynurenine pathway; and One-carbon metabolism primarily involving Folate cycle, Methionine cycle, Transsulfuration pathway, Polyamine pathway and Nucleotide biosynthesis showing 16 upregulated and 4 downregulated transcripts. Among the downregulated transcripts, 3 were observed in Cysteine synthase/cystathionine beta-synthase. Genes involved in polyamine synthesis viz., *odc-1* (ornithine decarboxylase) and *spds-1* (spermidine synthase) were upregulated which upon validation through RT-PCR showed up to 6 folds (after 24 h) and 14 folds (after 36 h) increase, respectively (Fig 8).

## 2.10 Validation of putative differentially expressed genes by qRT-PCR

A subset of transcripts that are differentially expressed during anhydrobiotic stress of *H.indica* was created. In total, 9 genes were chosen for expression profiling based on the earlier reports. The genes and their expression pattern with respect to FPKM value is represented in Fig 8. The genes involved in glyoxylate pathway, polyamine synthesis, heat shock proteins (*hsp*), citrate synthase and elongase genes were validated using quantitative realtime-PCR at 0 h, 24 h, 36 h and 48 h of anhydrobiotic induction. Out of the 3 hsp's, *hsp70* and *hsp60* showed significant up regulation at 36 h of desiccation with ~20- and 10-fold changes, while *hsp1* was

**Table 1. Differentially expressed desiccation stress response transcripts identified in anhydrobiotic *H. indica*.**

| Stress related pathway | No. of transcripts (Upregulated) | No. of transcripts (Downregulated) |
|---|---|---|
| **Antioxidative defense** | | |
| Superoxide dismutases (Mn) | 3 | - |
| Superoxide dismutases (Cu-Zn) | 1 | - |
| Glutathione peroxidase | 1 | - |
| Glutathione s-transferase (gst) | 2 | - |
| Glucose-6-phosphate dehydrogenase | 1 | - |
| Peroxiredoxins (1-cys peroxiredoxin) | 3 (2) | - |
| Thioredoxins reductase | 1 | - |
| Thioredoxin peroxidase (TPx) | 1 | - |
| Peroxisomal biogenesis factors (pex) | 2 | - |
| **Detoxification pathway** | | |
| ABC transporters | 11 | 1 |
| Multi-drug resistance protein (mrp/cdr) | | |
| *mrp-8* | 2 | - |
| *cdr-1* | 1 | - |
| Other ABC transporters | 9 | 1 |
| Short chain dehydrogenase/reductase (SDRs) | 2 | 2 |
| Nuclear hormone receptors (NHRs) | 2 | - |
| UDP-glucuronosyltransferase | - | 2 |
| **Signal transduction** | | |
| **Small GTPases** | 20 | 1 |
| ARF | 5 | 1 |
| Rho | 12 | - |
| Rab | 3 | - |
| Ran | 1 | - |
| **Others** | | |
| *plc-1* | 2 | - |
| Protein Kinase C | 2 | - |
| Diacylglycerol kinase | 2 | |
| Calcium ion binding protein | 1 | |
| MAPK/P38 | 1 | - |
| **Other Kinases** | 45 | 8 |
| **Unfolded protein response and Molecular Chaperones** | | |
| TCP-1 chaperonin family | 9 | - |
| *dnj-13* | 1 | - |
| ***peptidyl-prolyl cis-trans isomerase*** | | |
| FKBP type | 1 | - |
| Cyclophilin type | 6 | - |
| protein disulfide isomerase | 1 | 1 |
| *gfat-1* | 1 | - |
| alpha-1,2-mannosidase | 1 | - |
| *nsf-1* | 1 | - |
| *pqn-95* | 3 | - |
| signal recognition particle | 3 | - |
| protein glycosylation | 7 | - |
| ***Heat shock proteins*** | | |

*(Continued)*

**Table 1.** (Continued)

| Stress related pathway | No. of transcripts (Upregulated) | No. of transcripts (Downregulated) |
|---|---|---|
| *hsp60* family | 1 | - |
| *hsp70* family | 4 | - |
| **Ubiquitin-proteasome system** | | |
| Serine-type carboxypeptidase | 2 | - |
| Vacuolar ATPase (*vha-4*, *vha-14*) | 2 | - |
| Cathepsin D-like aspartic protease | 1 | - |
| ***20S core proteasome core*** | | |
| Proteasome subunit alpha type (*pas-1*) | 1 | - |
| Proteasome subunit alpha type (*pas-2*) | 1 | - |
| Proteasome subunit alpha type (*pas-3*) | 1 | - |
| Proteasome subunit alpha type (*pas-4*) | 1 | - |
| Proteasome subunit alpha type | 1 | - |
| Proteasome subunit beta type (*pbs-2*) | 1 | - |
| Proteasome subunit beta type | 3 | - |
| ***19S proteasome regulatory particle*** | | |
| *rpn-1* | 1 | - |
| *rpt-5* | 1 | - |
| Others | 4 | - |
| ***Ubiquitin tagging enzymes*** | | |
| ubiquitin-activating enzyme (E1) | 1 | - |
| ubiquitin-conjugating enzyme (E2) | 2 | - |
| ubiquitin ligases (E3) | 1 | - |
| ***Protein sorting*** | | |
| AAA ATPase | 4 | - |
| *cdc-48.2* | 1 | - |
| Cation transport ATPase | 3 | - |

significantly up regulated at both 36 h and 48 h. Two genes involved in polyamine synthesis, *odc-1* (ornithine decarboxylase) and *spds-1* (spermidine synthase) were upregulated up to 6 folds (24 h) and 14 folds (36 h), respectively. Genes encoding isocitrate lyase, malate synthase and citrate synthase involved in the glyoxalate pathway, showed upregulation of more than 20, 60 folds and >60 folds, respectively, at 36 h. The elongase gene *elo-7* showed ~45-fold changes at 36h of desiccation (Fig 9).

**Table 2. Differentially expressed repeat protein class transcripts identified in anhydrobiotic *H. indica*.**

| Repeat protein | No. of transcripts (Upregulated) | No. of transcripts (Downregulated) |
|---|---|---|
| HEAT_class | 4 | - |
| WD-40 class | 2 | - |
| Repeating small protein class | 1 | - |
| Tetratricopeptide class | 1 | - |
| Ankyrin | 2 | - |
| Spectrin repeat-containing domain protein | 4 | - |
| M protein | 1 | - |

**Table 3. Differentially expressed transcripts encoding key enzymes of metabolic pathways altered in anhydrobiotic *H. indica*.**

| Altered metabolic pathways | No. of transcripts upregulated | No. of transcripts downregulated |
|---|:---:|:---:|
| **Fatty acid** | | |
| **Fatty acid biosynthesis** | 5 | 1 |
| elo-7* | 1 | - |
| **Production of free fatty acid** | 3 | - |
| **Gluconeogenesis** | | |
| **Beta-oxidation of fatty acids** | | |
| Lipase | 2 | - |
| Acyl-CoA dehydrogenase | 1 | - |
| Ketothiolase | 2 | - |
| Carnitine acetyltransferase | 2 | - |
| **Glyoxylate pathway** | | |
| Citrate synthase* | 3 | - |
| Bifunctional enzyme (isocitrate lyase / malate synthase) * | 7 | - |
| Aconitase | 1 | - |
| Malate dehydrogenase | 3 | - |
| **Glyceroneogenesis** | | |
| Glutamate dehydrogenase | 1 | - |
| Phosphoenolpyruvate carboxykinase—PEPCK (ATP binding) | 1 | - |
| Phosphoglycerate dehydrogenase | 1 | - |
| Glycerol-3-phosphate dehydrogenase | 3 | - |
| **Mevalonate pathway** | | |
| HMG-CoA reductase | 1 | - |
| Diphosphomevalonate decarboxylase | 1 | - |
| Squalene synthase | 1 | - |
| **Glyceraldehyde-3-phosphate dehydrogenase** | | |
| Glyceraldehyde-3-phosphate dehydrogenase | 1 | 4 |
| **Amino-acid metabolism** | | |
| **Lysine biosynthesis** | | |
| Aspartate semialdehyde dehydrogenase | 1 | - |
| **Saccharopine pathway** | | |
| Saccharopine dehydrogenase (SDH) | 1 | - |
| Pyrroline-5-carboxylate reductase (P5CR) | 1 | - |
| **Shikimate pathway** | | |
| DAHP synthase | 2 | - |
| Chorismate synthase | 1 | |
| Anthranilate synthase | 1 | - |
| Tryptophan synthase | 1 | - |
| **Kynurenine pathway** | | |
| Tryptophan 2,3-dioxygenase | 1 | - |
| Quinolinate phosphoribosyltransferase | 1 | - |
| **One-carbon metabollism** | | |
| **Folate cycle** | | |
| Serine hydroxymethyltransferase | 2 | - |
| Methylene tetrahydro folate dehydrogenases | 1 | - |
| **Methionine cycle** | | |
| S-adenosylmethionine-dependent methyltransferase | 1 | - |
| S-adenosylmethionine synthase | 1 | - |

(*Continued*)

**Table 3.** (Continued)

| Altered metabolic pathways | No. of transcripts upregulated | No. of transcripts downregulated |
|---|---|---|
| S-adenosylhomocysteine hydrolase | 1 | - |
| **Transsulfuration pathway** | | |
| glutathione-disulfide reductase | 1 | - |
| hydroxyacylglutathione hydrolase | 1 | - |
| S-(hydroxymethyl) glutathione dehydrogenase | 1 | - |
| Cysteine synthase/cystathionine beta-synthase | - | 3 |
| cystathionine γ- lyase | 1 | - |
| **Polyamine pathway** | | |
| ornithine decarboxylase* | 1 | - |
| sperimidine synthase* | 1 | - |
| **Nucleotide biosynthesis** | | |
| phosphoribosylglycinamide formyltransferase | 1 | - |
| phosphoribosylaminoimidazolecarboxamide formyltransferase | 2 | - |
| Nucleoside diphosphate kinase | 1 | 1 |

* The transcripts encoding these enzymes were validated using real-time PCR

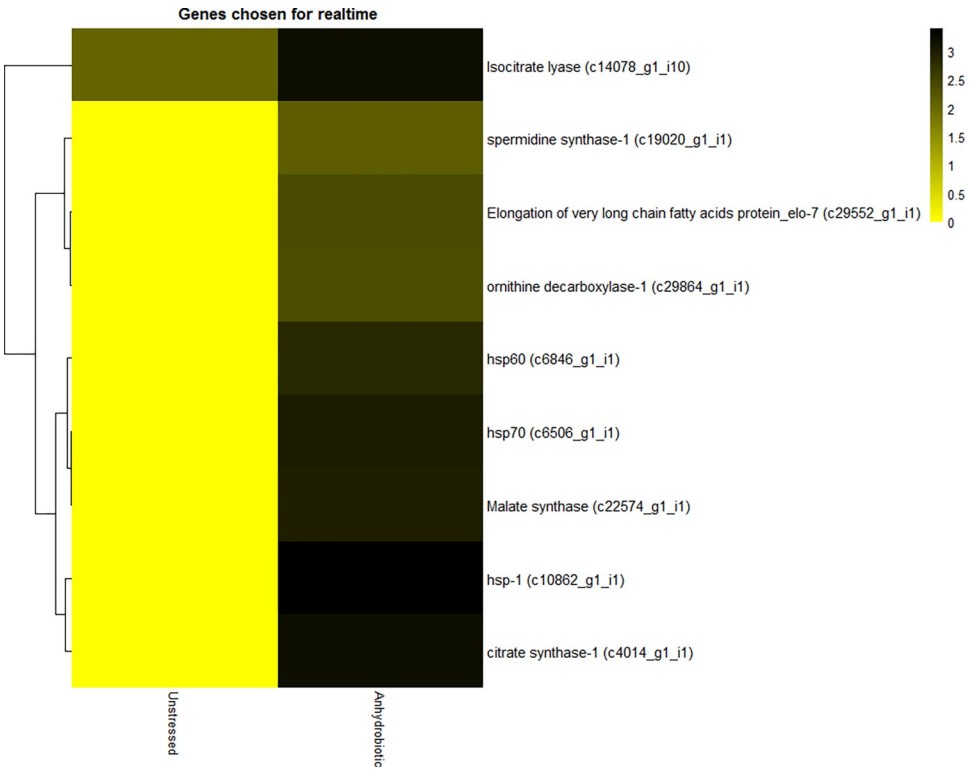

**Fig 8. Expression profile of genes selected for qRT-PCR study based on FPKM value (values are log transformed).**

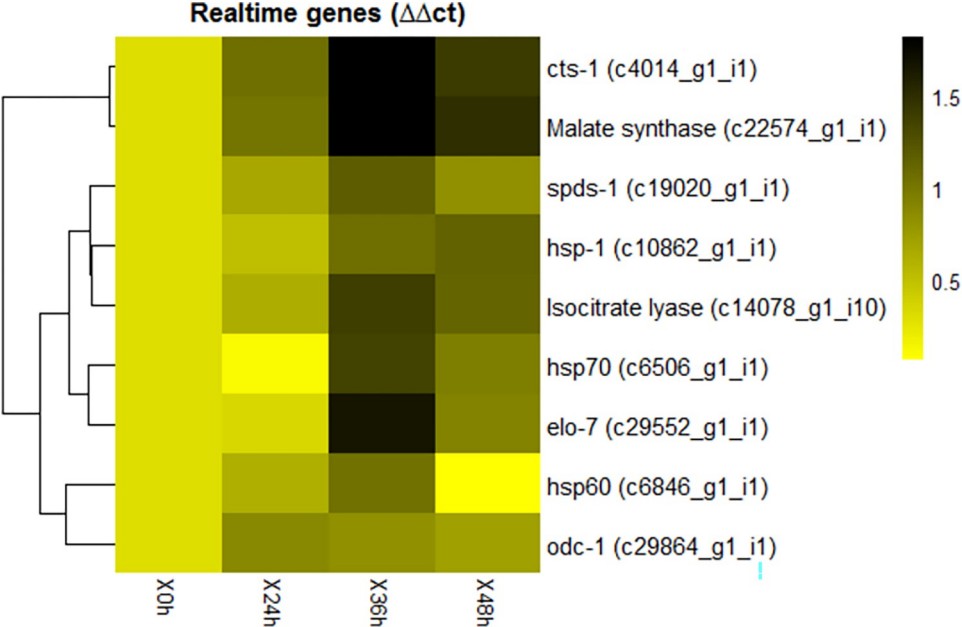

**Fig 9. Temporal expression pattern of genes during induction of anhydrobiosis in *H. indica* validated using qRT-PCR.**

## 3. Discussion

### 3.1 Molecular alterations in *H. indica* to overcome desiccation stress

**3.1.1 Antioxidant defense.** Reactive oxygen species (ROS) is a major constraint during anhydrobiosis due to its damaging effects in the biomolecules [24]. The antioxidant defense strategy is universal among anhydrobiotes as they act as scavengers to the ROS which is mediated by a repertoire of genes [28]. Upregulation of antioxidant enzymes in anhydrobiotic *H. indica* are in conformation with previous reports on cryptobiotic midge *Polypedilum vanderplanki*, anhydrobiotic nematode *Panagrolaimus superbus* and cryptobiotic tartigrades *Echiniscoides* cf. *sigismundi* and *Richtersius* cf. *coronifer* [17, 29, 30]. Higher FPKM value of DNA damage-inducible protein-1 in anhydrobiotic IJs signifies its importance in desiccation stress tolerance as they are reported to play a crucial role in DNA repair pathways in *C. elegans* [31–32] and *Arabidopsis* [33].

**3.1.2 Detoxification mechanisms for anhydrobiotic survival.** The detoxification enzymes eliminate toxic products through membrane transporters like ATP-binding cassette [34]. Unlike as reported in anhydrobiotic nematodes *Plectus murrayi*, *C. elegans* and *Aphelenchoides fragariae* [18, 24, 34], the Phase II detoxification enzyme UGT was downregulated in anhydrobiotic *H.indica*. Since the detoxification is a metabolically exorbitant process, we speculate that to compensate the UGT downregulation, anhydrobiotic *H. indica* recruits the other Phase II enzyme GST with a dual role in the antioxidant defense and detoxification process, as evident from its upregulation.

**3.1.3 Signal transduction in anhydrobiotic stress response.** The onset of stress activates signal transduction pathways generating secondary messengers like **ROS**, calcium, and inositol phosphates which activate stress transcription factors and responsive genes through kinases/phosphatases [35]. As in the anhydrobiotic *H. indica* the genes involved in phosphatidylinositol signaling were also essentially responsible for dehydration stress tolerance in red seaweed *Gloiopeltis furcate* [36] and several other stresses in higher plants [35]. Likewise, the role of *plc-*

*1* was crucial in osmotic stress tolerance [37] and cell growth in *S. cerevisiae* [38] and regulating ovulation in *C. elegans* [39]. Upregulation of MAPK/p38, known to transduce the stress signals to protective genes, has also been examined in few other nematodes viz., *C. elegans*, *Aphelenchus avenae*, and *Panagrolaimus superbus* [40] while small GTPases, regulating JNK and p38 MAPKs in *C. elegans* [41], and stress response mechanisms in plants [42–44]. Upregulation of a single transcript encoding casein kinase II was also reported in another EPN *Steinernema feltiae* [11] and anhydrobiotic free-living nematode *P. superbus* [29].

**3.1.4 Unfolded protein response and molecular chaperones.**   The unfolded protein response (UPR) is a cell-signaling system that readjusts the endoplasmic reticulum (ER) folding capacity to restore protein homeostasis [45–47]. Anhydrobiotes are equipped with an arsenal of molecular chaperones to mitigate the ER stress through ER stress response or UPR. The heat shock proteins (HSPs) prevent misfolding of protein and aggregation during moisture stress in nematodes [18, 24, 29, 34]. Upregulated single transcript of *Dnj-13* in anhydrobiotic *H. indica*, which is a class B J-protein belonging to *hsp40* family and a co-chaperone of *hsp70* is also an ortholog of human DNAJB 1 working synergistically in human and nematode HSP70 systems, and exhibits efficient protein disaggregation activity [48]. In *C. elegans* the FKBP assists in cold-sensitivity and secretion of the extracellular matrix [49], while in *Scenedesmus sp.*, a thermophilic microalga, it is responsible for thermal and salinity stress tolerance [50]. Its upregulation in anhydrobiotic *H. indica* may have an analogous role in the moisture stress tolerance.

Upregulation of PDI which helps to catalyze the formation and rearrangement of cysteine residues and post-translational modification for proper folding of protein [46, 48] was also reported from the free-living anhydrobiotic nematode, *P. superbus* [29]. We also report a downregulated PDI transcript. PDI attenuates the UPR signaling by acting as a modulator IRE-1, an activator of UPR signaling [45]. Thus, it can be concluded that the upregulated PDI transcripts mediate proper protein folding while the downregulated might assist in maintenance of UPR signaling during prolonged stress conditions. Human NSF (N-ethylmaleimide sensitive factor) is predicted to be involved in IRE-1 mediated UPR [51], thus, upregulation of Nsf 1 in anhydrobiotic *H. indica* supports our hypothesis of protein homeostasis in anhydrobiotic IJ.

Upregulation of the signal recognition particle (SRP) which recognizes the newly formed nascent polypeptides and enables its insertion into ER, alongside microtubule forming genes T-complex protein 1, α, and β-tubulin in anhydrobiotic *H. indica* have been reported in anhydrobiotic *P. superbus* [29]. Interestingly, the upregulation of UPR genes alpha 1, 2- mannosidase and *gfat-1* in anhydrobiotic *H. indica*, which parasitizes insect, have also been reported from anhydrobiotic plant-parasitic nematode *A. fragariae* [34]. Three upregulated *pqn-95* were found to be homologous to *C. elegans abu-14* (activated in blocked UPR) protein assisting its innate immunity [52]. Protein glycosylation and ER-associated functions are essential for cellular homeostasis during desiccation stress [53] which has been corroborated by the upregulation of transcripts involved in protein glycosylation, glycoprotein quality control and protein folding in anhydrobiotic *H. indica*.

**3.1.5 Misfolded protein degradation by ubiquitin-proteasome system.**   The 'ubiquitin-proteasome system' (UPS) is the major selective protein degradation pathway among eukaryotes which degrades the damaged proteins and maintains protein homeostasis. Expression of several UPS genes in anhydrobiotic *H. indica* with found parity with another anhydrobiotic nematode *P. superbus* [29]. During the polyubiquitination process, the ubiquitin-binding factors like *cdc48* provide the necessary driving force to determine the direction of transport for retro-translocation [54, 55], also reported by us.

**3.1.6 Repeat proteins.** The importance of repeat proteins in nematode stress survival is poorly understood. However, the repeat proteins upregulated in our studies consisted of Tetratricopeptides, known to mediate protein interactions with partner proteins involved in plant stress and hormone signaling [56]. We also report WD40 having an important role in signal transduction in drought stress tolerance of *Arabidopsis* [57] and abiotic stress tolerance of *Triticum aestivum* [58]. It is quite evident that many of the repeat protein gene family members have a significant role in plant stress response and its upregulation in anhydrobiotic *H. indica* provides an insight on the evolutionary role for variable adaptations to moisture stress. This is the first report on the repertoire of repeat proteins and their expression in anhydrobiotic nematodes, though previously reported in unstressed *P. superbus* [29]. Two upregulated ankyrin repeats in anhydrobiotic *H. indica* also plays a role in *C. elegans* subjected to ionizing radiation [32].

## 3.2 Altered metabolism in anhydrobiotic *H. indica* to meet energy demand during desiccation stress

Gluconeogenesis is a preferred pathway in moisture stress tolerance wherein glucose produced is metabolized to stress-protectant trehalose [11]. The β-oxidation of fatty acid essentially provides substrates for gluconeogenesis via breakdown of non-carbohydrate precursors [59]. An increase in lipolysis followed by β-oxidation of fatty acids was reported in *C. elegans* during the onset of anhydrobiosis. Both, in *C. elegans* and unrelated yeast, the glyoxylate shunt (GS) enables the synthesis of trehalose from acetate, a critical factor for preconditioning to anhydrobiotic survival, while its inhibition dramatically decreases the potential to tolerate moisture stress [60]. The role of the GS has been reported in anhydrobiotic *A. avenae* [12], *Plectus murrayi* [24], dauer stage *C. elegans* [61], and *Romanomermis sp.*, [62]. All the 5 key enzymes of GS [59] showed significant upregulation in the anhydrobiotic *H. indica* IJ. The transcripts encoding glycogen synthase kinase 3 (*gsk-3*) and casein kinase 2 (*CK2*) were significant upregulated; which negatively regulate the expression of glycogen synthase gene, thus, inhibiting glycogen synthesis [63]; while *gsk-3* positively regulate the genes encoding gluconeogenic enzymes PEPCK–a key enzyme in glyceroneogenesis and glucose-6-phosphatase [64]. Glyceroneogenesis is deployed to produce glycerol as stress protectant [65–67]; or to produce triglyceride possibly to participate in gluconeogenesis [11]; or an important role in the synthesis of serine which is a key source of carbon in the one-carbon metabolism [68] from non-glucose precursors like glutamine [65]. Transcripts encoding the glutamate dehydrogenase and PEPCK were significantly upregulated in anhydrobiotic IJ, wherein, the former produces oxaloacetate from glutamine while the latter produces phosphoenolpyruvate from oxaloacetate. The transcript encoding NAD$^+$-dependent glutamate dehydrogenase showed upregulation. In yeast, it catalyzes the conversion of L-glutamate to alpha-ketoglutarate, while NADP dependent catalyzes the synthesis of glutamate from alpha-ketoglutarate [66]. The *H. indica* may deploy glyceroneogenesis to produce glycerol-3-phosphate which then dephosphorylated into glycerol by *gpdh*. Stress-induced expression of *gpdh* has been reported in *Osmerus mordax*, *S. mansoni*, *C. elegans* [68] and yeast [69].

We observed a significant upregulation in the transcript encoding HMG-CoA reductase, a rate-limiting enzyme in the isoprenoid/mevalonate pathway, that produces sterols [70], and squalene synthase gene to help catalyze the initial step in cholesterol synthesis [71], although the squalene synthase gene was reportedly absent in *C. elegans* [72]. The role of mevalonate pathway and cholesterol in the moisture stress response of *H. indica* needs further investigation as the nematode is in a symbiotic association with the bacterium *Photorhabdus luminescens*.

The Glyceraldehyde-3-phosphate dehydrogenase (GAPDH) functions as a metabolic switch during oxidative stress to generate more NADPH to aid NADPH-dependant antioxidant enzymes like thioredoxin and glutaredoxin [73], alongside a few other non-metabolic roles [74–76]. In *C. elegans* the role of GAPDH in hypoxia or anoxia stress tolerance and survival has been reported [77]. Our studies similarly reveal differential expression of transcripts encoding GAPDH which suggests that *H. indica*, as an evolutionary backup, switches off its metabolic role and recruits GAPDH to take up non-metabolic roles.

The Saccharopine pathway catabolizes the amino acid lysine into proline and pipecolate, enabling the plants to tackle abiotic stress [78–80], while supplementation of proline and tryptophan increased thermotolerance in *C. elegans* [81]. A transcript encoding aspartate semialdehyde dehydrogenase showed significant upregulation in anhydrobiotic *H. indica*, suggesting that the nematode may deploy the Aspartate pathway for lysine biosynthesis to produce stress protectants proline and pipecolate through Saccharopine pathway. To best of our knowledge, this is the first report of the upregulation of Saccharopine pathway in moisture stress tolerance of *H. indica*.

The Shikimate pathway is present across bacteria, fungi, plants but not in animals, who obtain the aromatic amino acids essentially from their diet and nutritional support from symbiotic organisms [82]. Interestingly, we observed significant upregulation of 2 transcripts that encode for 3-deoxy-d-arabino-heptulosonate 7-phosphate synthase (DAHP) in anhydrobiotic *H. indica* which initiates the Shikimate pathway. However, there was no significant upregulation in the transcript homologous to *aro-1*, which catalyzes the intermediate steps of this pathway [83]. The chorismate produced via Shikimate may act as a precursor for tryptophan biosynthesis. The first step in the biosynthesis of tryptophan from chorismate is catalyzed by anthranilate synthase and tryptophan synthase which were upregulated in anhydrobiotic *H. indica*. Probably this is the first report on the role of Shikimate pathway genes in moisture stress tolerance. To check the presence of Shikimate pathway genes in nematodes, a standalone blast was performed. Of all the available EPN genomes, we found homolog of DAHP synthase transcript only in *S. glaseri*, but not in the model nematode *C. elegans*. Since *H. indica* is symbiotically associated with *P. luminescens*, it might have acquired it through horizontal gene transfer; however, revalidation is essential to confirm the presence of Shikimate pathway genes in this nematode. Transcripts encoding TDO and quinolinate phosphoribosyltransferase (QRPT) involved in the kynurenine pathway which catabolizes tryptophan into $NAD^+$ was upregulated in anhydrobiotic *H. indica*, as also reported in other animals [84].

Polyamines (spermidine and putrescine) are essential for desiccation tolerance in dauer *C. elegans as* the *spds-1* and *odc-1* mutants were extremely sensitive to moisture stress [18]. Similar observation was made in anhydrobiotic *H. indica*. Glutathione (GSH) is biosynthesized from its precursor cysteine through homocysteine degradation in the transsulfuration pathway acting as antioxidant to maintain redox homeostasis [28, 85]. Detoxification process in anhydrobiotic *H. indica* was assisted via upregulated transcripts encoding hydroxyacylglutathione hydrolase/glyoxalase II, and S-(hydroxymethyl) glutathione dehydrogenase, wherein, the former is part of the methylglyoxal detoxification system and the latter a part of the formaldehyde detoxification system [86, 87].

The EPN *H. indica* along with its symbiotic bacteria *P. luminescens* is a golden fleece to manage the crop pests' pandora box. This nematode possesses remarkable insecticidal potential, with significant commercial success globally, based on its unique ability to survive moisture stress. Our study has provided crucial insights on many unknown molecular events adopted by this nematode to withstand unfavorable environment. This information can provide critical inputs for developing stable, durable and effective products. The results also open-up new avenues to understand and inter-link the process of desiccation stress tolerance among unrelated organisms.

## 4. Materials and methods

### 4.1 Rearing and induction of anhydrobiosis in the infective juveniles

The EPN *Heterorhabditis indica* strain KX601067 was originally isolated from the agricultural farms of IARI, New Delhi [88] and routinely cultured *in vivo* on the Greater wax moth, *Galleria mellonella*, larvae [89], using the White's Trap [90]. Anhydrobiosis was induced by incubating the freshly emerged IJs for two days in 97% relative humidity chambers maintained by means of water-glycerol solution [91, 92]. Approximately 1x10⁵ IJs of *H. indica* concentrated in sterile distilled water were slowly released, in the centre of a 2 cm diameter Qualitative Filter paper disc (Grade 1; Retention 2.5 μm; HIMEDIA 6010-900-100C) using a micro-pipette. The discs were placed on a wire-mesh which was suspended and held at 3 cm from the top of a 500 ml glass beaker. The beaker was filled with 300 ml glycerine-water solution (10.32: 89.68 ml ratio). The beaker was sealed airtight and kept at 27˚C for 48 h. Thereafter, the discs were removed which contained the anhydrobiotic IJs of *H. indica* (S2 Plate). For validation of genes by RT-PCR, the IJs were removed from the experimental set-up at 3 time-lines (24h, 36h, 48h) during the course of inducing anhydrobiosis with 0 h (unstressed IJ) serving as control. The IJ collected at different time-lines were flash frozen in liquid nitrogen and stored at -80˚C.

### 4.2 RNA isolation, RNA-seq and bioinformatics data analysis

**4.2.1 Isolation of total RNA and RNA-Seq.** The IJ of *H. indica* were subjected to 97% relative humidity chamber for 0h (unstressed) and 48h (anhydrobiotic), and samples (approximately 3,000 IJ) were taken from five independent sets for each treatment and flash frozen in liquid nitrogen [10, 21, 93]. Total RNA was isolated from each pooled sample (approximately 15,000 IJs) using TRIzol reagent (Invitrogen, USA) as per manufacturer's instructions. The isolated total RNA was treated with DNaseI to remove any traces of DNA. Quality and quantity of isolated RNA was measured using NanoDrop-1000 and agarose gel electrophoresis (1.2%). Libraries were produced by generating cDNA from RNA and then adding adapters to the cDNAs and sequenced on Illumina HiSeq 2500 platform. The sequencing depth is ~50x and the resulted raw paired-end sequences were subjected to standard quality filtering procedures such as removal of adapter sequences; removal of poor quality reads based on phred score etc. The raw sequence data has been submitted in GenBank NCBI under BioProject ID. PRJNA789679, Bio sample No. SAMN24146495 and SRA IDs. SRR17284459, SRR17284460.

**4.2.2 Bioinformatics data analysis.** *4.2.2.1 De novo transcriptome assembly*. The fastq files were trimmed using *Cutadapt* [94] to remove adapter sequences before performing the assembly. The fastq files were subjected to quality check before and after adapter removal, and the poor quality reads were filtered and assembled using *Trinity* [95] with default options. Transcripts of length ≥ 200 bp were focused for expression, estimation and downstream annotations. *Assembly stats* of *Trinity* was used to carry out a comparative study between our assembled data with Somvanshi *et al.* (2016) [26] and Bhat *et al.*, (2022) [27] in terms of the quality check. We also carried out an alignment using *Bowtie 2* [96] and *HISAT 2* [97], and reference-based mapping using *Gmap* [98].

*4.2.2.2 Transcriptome annotation*. The assembled transcripts were annotated using in-house pipeline of SciGenom Labs Pvt Ltd. CANoPI (CANoPI–Contig Annotator Pipeline). Briefly, the assembled transcripts were compared with NCBI non-redundant protein database using BLASTX program. Matches with E-value ≤ 10⁻⁵ and similarity score ≥ 40% were retained for further annotation. The top BLASTX hit of each transcript was studied and the organism name was extracted. The predicted proteins from BLASTX were annotated against NCBI, UniProt, Pathway and other databases. The gene ontology (GO) terms for transcripts

were extracted wherever possible. The total numbers of different GO terms were identified with respect to molecular function, biological process and cellular component category. The differentially expressed transcripts have been subjected to gene enrichment analysis using the Blast2GO software with filtering parameters set to FDR value <0.05 [99].

**ORF prediction.** *TransDecoder* [100] bioinformatics was used with default options to predict longest Open Reading Frames (ORFs) and amino acid sequences from the assembled transcripts.

**Transcriptome completeness.** *TransRate* [101] was used for assessing the quality of transcriptome and *CEGMA* (Core Eukaryotic Genes Mapping Approach [102]) and *BUSCO* (Benchmarking Universal Single-Copy Orthologs [103]) to evaluate the completeness of the assembly.

*4.2.2.3 Differential expression analysis.* The trimmed reads from unstressed and anhydrobiotic IJ were aligned to the assembled transcriptome using *Bowtie 2* program [96]. Up to 1-mismatch in the seed region (length = 31bp) and all multiple mapped position were reported. Differential gene expression analysis was performed using *DESeq* Bioconductor Package [104]. Transcripts having read count $\geq 1$ for both samples were chosen for differential expression analysis. Transcripts showing two folds change with p-value <0.01 were considered as differentially expressed.

**Heatmaps.** Heatmaps were generated using *pheatmap* [105] an R package for—i) transcripts chosen for validation using quantitative-PCR, and ii) anhydrobiosis specific pathway-wise transcripts. The heatmaps were generated based on log-transformed FPKM value.

## 4.3 Validation of transcripts using quantitative-PCR analysis

The transcripts involved in molecular chaperone activity of *hsp 70*, *hsp 60*, and *hsp 1*; metabolic pathways viz. glyoxylate pathway, polyamine biosynthesis pathway, fatty acid biosynthesis

**Table 4. List of primers used for the differential expression analysis through Real Time PCR.**

| Sl. No. | Transcript id | Gene name | Sequence |
|---------|---------------|-----------|----------|
| 1 | c6506_g1_i1 | *hsp70* | GTATGATGACGGTTACGACTCC-f |
| | | | TGCCAAATAACGGCCTCTAATA-r |
| 2 | c6846_g1_i1 | *hsp60* | TCTACGATGAGCCTGAGTTCTA-f |
| | | | ACCTTGAATGGGTCGATGATAC-r |
| 3 | c10862_g1_i1 | *hsp1* | TGACGAGACCATTGCTTGG-f |
| | | | GAGATGATTGGGTTGGCCTTA-r |
| 4 | c19020_g1_i1 | *spds1* | CTCTAAGAACCCAAACGTCACT-f |
| | | | CGTGGATCTGCTTGTTGTAGTA-r |
| 5 | c29864_g1_i1 | *odc-1* | GGCGAATCACAAGGGAAGAA-f |
| | | | CAACATCGTGCGTCAAATGG-r |
| 6 | c14078_g1_i10 | Isocitrate lyase | CAGCAGTGAGGATGAGGAAATA-f |
| | | | TGATGAATCGGAGAGCATCAG-r |
| 7 | c22574_g1_i1 | Malate synthase | AAGTCTGCTAAGGCTGGTAAC-f |
| | | | GGTGGTCACATCGTCGTAAA-r |
| 8 | c4014_g1_i1 | Citrate synthase | GAGAAAGACCGACCCAAGATAC-f |
| | | | AGCGACCTCGTAGATGTTAGA-r |
| 9 | c29552_g1_i1 | *elo-7* | GACACTACGGTGAGAGCTAGAA-f |
| | | | GCCGCTTCGTTGCTTAGATTA-r |
| 10 | 18srRNA | Reference gene | CTGCATAGCAGATCCAGTGATT-f |
| | | | CCCATGAGGGTAGAGCATAGA-r |

pathway were chosen for validation using quantitative-PCR based on earlier reports [17, 18, 24, 29, 106]. The RNA was isolated from the IJ exposed to relative humidity chambers for 0h, 24h, 36h and 48h IJs using TriZol method as mentioned in 4.2. Further, one µg of DNase I (NEB,USA) treated RNA was used for the first strand cDNA synthesis using superscript III Reverse transcriptase (Invitrogen) kit. Quantitative Real Time PCR was done in Step One Plus instrument (ABI) using three biological and three technical replicates for each treatment. Each reaction contained 5 µl 2X SYBR Master mix reagent (Takara), 1µL cDNA and 400 nM of gene specific primers in a final volume of 10 µl. Each pair of primers was designed using *Primer-Quest* tool of IDT with an amplicon size of 100–130 bp. The specificity of reaction was analysed in melting curve analysis. The relative transcript level of the mRNA was determined by $\Delta\Delta^{CT}$ values in comparison with unstressed IJ and 18S rRNA gene was used as internal reference gene [21, 93]. For the analysis at different time points, Ct values of control was taken as one and fold change in other time points were calculated by similar method. The primers used for the expression analysis of *H. indica* genes are given below (Table 4):

## Supporting information

**S1 Plate. Metabolic pathways altered in anhydrobiotic *H. indica*.**
(TIF)

**S2 Plate.** *Heterorhabditis indica* infective stage juveniles (a) unstressed (b) anhydrobiotic (x20).
(TIF)

**S1 File. Gene ontology for whole transcriptome of *Heterorhabditis indica*.**
(XLS)

**S2 File. Gene ontology for differentially expressed transcripts of anhydrobiotic *Heterorhabditis indica* against unstressed IJ.**
(XLS)

**S3 File. Differentially expressed desiccation stress response transcripts identified in anhydrobiotic *Heterorhabditis indica*.**
(XLSX)

**S4 File. Transcripts of different biochemical pathways that are differentially expressed during desiccation stress in anhydrobiotic *Heterorhabditis indica*.**
(XLSX)

**S1 Fig. Differential expression of Anti-oxidant defense related genes identified in anhydrobiotic *H. indica* (values are log transformed).**
(TIF)

**S2 Fig. Differential expression of detoxification system related genes identified in anhydrobiotic *H. indica* (values are log transformed).**
(TIF)

**S3 Fig. Differential expression of signal transduction related genes identified in anhydrobiotic *H. indica* (values are log transformed).**
(TIF)

**S4 Fig. Differential expression of unfolded protein response related genes identified in anhydrobiotic *H. indica* (values are log transformed).**
(TIF)

**S5 Fig. Differential expression of ubiquitin-proteasome related genes identified in anhydrobiotic *H. indica* (values are log transformed).**
(TIF)

**S6 Fig. Differential expression of repeat protein class genes identified in anhydrobiotic *H. indica* (values are log transformed).**
(TIF)

**S7 Fig. Differential expression of β-oxidation of fatty acids genes identified in anhydrobiotic *H. indica* (values are log transformed).**
(TIF)

**S8 Fig. Differential expression of glyoxylate pathway genes identified in anhydrobiotic *H. indica* (values are log transformed).**
(TIF)

**S9 Fig. Differential expression in genes of glyceroneogenesis identified in anhydrobiotic *H. indica* (values are log transformed).**
(TIF)

**S10 Fig. Differential expression in genes of mevalonate pathway identified in anhydrobiotic *H. indica* (values are log transformed).**
(TIF)

**S11 Fig. Differential expression in genes of mevalonate pathway identified in anhydrobiotic *H. indica* (values are log transformed).**
(TIF)

**S12 Fig. Differential expression in amino acid metabolism related genes identified in anhydrobiotic *H. indica* (values are log transformed).**
(TIF)

**S13 Fig. Differential expression in one-carbon metabolism related genes identified in anhydrobiotic *H. indica* (values are log transformed).**
(TIF)

**S1 Table.** A. Raw read summary. B Trimmed read summary.
(DOCX)

**S2 Table. Assembled transcript summary of *H. indica* transcriptome.**
(DOCX)

**S3 Table. *TransRate* results of *H. indica* transcriptome assembly.**
(DOCX)

**S4 Table. CEGMA results of *H. indica* transcriptome assembly.**
(DOCX)

**S5 Table.** A. BUSCO results of *H. indica* transcriptome against Eukaryota protein sets. B. BUSCO results of *H. indica* transcriptome against Nematoda protein sets.
(DOCX)

**S6 Table. Comparative assemble stats report of fresh + anhydrobiotic *H. indica* and Somvanshi *et. al.*, (2016).**
(DOCX)

**S7 Table. Read alignment and expression summary of *H. indica* transcriptome.**
(DOCX)

**S8 Table. Transcript expression and total number of transcripts (length $\geq$ 200bp) of *H. indica*.**
(DOCX)

**S9 Table. Gene enrichment analysis of differentially expressed genes in anhydrobiotic *H. indica*.**
(DOCX)

# Acknowledgments

Authors thank Viswanathan Sateesh, Kommu Kiran Kumar for assisting in nematode culturing and transcriptomics; and Ramya for preparing biochemical chart and drafting the manuscript.

# Author Contributions

**Conceptualization:** Manimaran Balakumaran, Anil Sirohi, Kishore Gaikwad, Sharad Mohan.

**Data curation:** Manimaran Balakumaran, Atmakuri Ramakrishna Rao, Sharad Mohan.

**Formal analysis:** Manimaran Balakumaran, Yuvaraj Iyyappan, Atmakuri Ramakrishna Rao, Sarika Sahu, Sharad Mohan.

**Funding acquisition:** Manimaran Balakumaran.

**Investigation:** Manimaran Balakumaran, Parameshwaran Chidambaranathan, Sharad Mohan.

**Methodology:** Manimaran Balakumaran, Parameshwaran Chidambaranathan, Jagannadham Prasanth Tej Kumar J. P., Anil Sirohi, Sharad Mohan.

**Project administration:** Sharad Mohan.

**Resources:** Manimaran Balakumaran, Sharad Mohan.

**Supervision:** Sharad Mohan.

**Validation:** Manimaran Balakumaran, Sharad Mohan.

**Visualization:** Sharad Mohan.

**Writing – original draft:** Manimaran Balakumaran, Parameshwaran Chidambaranathan, Jagannadham Prasanth Tej Kumar J. P., Sharad Mohan.

**Writing – review & editing:** Manimaran Balakumaran, Anil Sirohi, Pradeep Kumar Jain, Kishore Gaikwad, Anil Dahuja, Sharad Mohan.

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
