## [Decision Letter · Decision Letter 0]

3 Mar 2022

PONE-D-21-38148Deciphering the mechanism of anhydrobiosis in the entomopathogenic nematode Heterorhabditis indica through comparative transcriptomicsPLOS ONE

Dear Dr. Mohan,

Thank you for submitting your manuscript to PLOS ONE. After careful consideration, we feel that it has merit but does not fully meet PLOS ONE’s publication criteria as it currently stands. Therefore, we invite you to submit a revised version of the manuscript that addresses the points raised during the review process.

We look forward to receiving your revised manuscript.

Kind regards,

Anil Kumar Singh, Ph.D.

Academic Editor

PLOS ONE

Journal Requirements:

2. Please upload a new copy of Figures 2, 4 and 5 as the detail is not clear. Please follow the link for more information: " ext-link-type="uri" xlink:type="simple">https://blogs.plos.org/plos/2019/06/looking-good-tips-for-creating-your-plos-figures-graphics/"
https://blogs.plos.org/plos/2019/06/looking-good-tips-for-creating-your-plos-figures-graphics.

 “This is part of PhD research carried out by the first author Mr Manimaran Balakumaran  funded by the Post Graduate School, ICAR-IARI, New Delhi, India - 110012”

Reviewers' comments:

Reviewer's Responses to Questions

**Comments to the Author**

1. Is the manuscript technically sound, and do the data support the conclusions?

Reviewer #1: Yes

Reviewer #2: Yes

2. Has the statistical analysis been performed appropriately and rigorously? 

Reviewer #1: Yes

Reviewer #2: N/A

3. Have the authors made all data underlying the findings in their manuscript fully available?

Reviewer #1: Yes

Reviewer #2: No

4. Is the manuscript presented in an intelligible fashion and written in standard English?

Reviewer #1: Yes

Reviewer #2: Yes

5. Review Comments to the Author

Reviewer #1: The manuscript “Deciphering the mechanism of anhydrobiosis in the entomopathogenic nematode Heterorhabditis indica through comparative transcriptomics” studied the comparative transcriptome analysis of unstressed and anhydrobiotic H. indica. The research paper comprehend the scientific information revealing the role of different metabolic pathways for survival of H. indica during anhydrobiosis. These kinds of novel work provide insight regarding anhydrobiotic survival strategies of H. indica which could be useful for designing novel EPN formulations of H. indica with enhanced and sustained shelf-life during field applications, enhancing its biological control potential. The manuscript is well written and has potential to be accepted in PLOS ONE with few points needs to be incorporated for improvement.

- The material and methods are well written, however what was the source of IJs of Heterorhabditis indica strain KX601067 and on what basis it was identified, is not mentioned.

- The authors have well documented the results and supported the discussion with relevant literature nicely. But, a concrete conclusion about application of this research is lacking. Add one or two lines of conclusion in light of future application of this research.

Reviewer #2: The authors have compared the relative transcript abundance by RNA-seq during anhydrobiosis in the entomopathogenic nematode Heterorhabditis indica. The infective juveniles of H.indica were subjected to drought stress and compared the transcriptome between the control and drought stressed stages. Since the dried formulations of EPN are used as an alternative to chemical pesticides, this study might actually help in designing longer lasting formulations. However, there are some concerns, which can be addressed by the authors so that the overall quality of the manuscript is better.

1. In line 86, authors have mentioned that the read length was between 30 and 81 bp. Authors need to elaborate more on it. The information on whether it was single end or paired end sequencing is also not provided in the Materials and methods section. It is highly unlikely that read lengths were between 30-81 bp. Such a small length will give a misassembled de novo transcriptome. Detailed information on the sequencing depth, number of reads, purity filtered reads and elimination criteria for shortlisting the purity filtered reads should be provided in the MM.

2. Though authors have randomised the experiment, there is no information on the number of biological replicates used in the study. In case a single replicate was used, authors should compare at least their assembly from control samples with the previously published transcriptome data (Somvanshi et al., 2016).

3. Since the draft genome of the H.indica has recently been published (Bhat et al., 2022), authors are encouraged to perform a reference based mapping and quantification to improve their data. Alternatively, they can also map the de novo assembled transcriptome on the genome and remove the false and chimeric assemblies to improve the overall quality.

4. The results are too short, whereas the discussion is too long. In fact discussion has portions of results. It is advised that results can be merged.

5. Some of the sections in MM can be merged, for e.g., 4.2.2.2 can be merged with 4.2.2.3 and 4.2.2.4. Similarly, 4.2.2.5 can be merged with 4.2.2.6

6. The data should be submitted to a public database and requisite information should be provided in the manuscript.

6. PLOS authors have the option to publish the peer review history of their article (what does this mean?). If published, this will include your full peer review and any attached files.

Reviewer #1: No

Reviewer #2: No

---

## [Author Response · Author response to Decision Letter 0]

30 May 2022

All the 5 points raised in the email by the Academic Editor has been addressed and included in the revision.

Individual comments from the Reviewers have been duly incorporated in the revision and the details are provided below:

Reviewer 1:

Response to comment 1: The EPN Heterorhabditis indica was isolated from the IARI farm, New Delhi (Mohan et al., 2016). The details are included in the revised manuscript.

Response to comment 2: As suggested the lines have been added at the end of the Discussion

Reviewer 2:

Response to comment 1: We have performed paired-end sequencing using Illumina Hi-Seq 2500 platform. The average sequence length of the trimmed reads was 81 bp. The details on the sequencing depth, number of reads, purity filtered reads and elimination criteria for shortlisting the purity filtered reads are included in the revised manuscript (Supplementary Table 1 a,b)

Response to comment 2: Our analysis comprised of pooled samples from three independent biological replicates using 5000 IJs for each. Further, transcriptome library comparative assembly statistics using (assembly stats of Trinity) reveal that our transcriptome library statistics are relatively better than the transcriptome assembly of Somvanshi et al. (2016) by the following means: For example, N50 value of ours is 2843bp while it is 1291bp. Similarly, other parameters like N60, N70, N80, N90 and N100 are also higher in our library. Also, our sequence is clean as there are no ambiguous bases, whereas, in Somvanshi et al. (2016) it is 1670. Our assembly does not have any gaps, while it is 66 in number for them. So comparatively our assembly is superior in many ways compared to theirs. The details are included in Supplementary Table 6 in the manuscript. 

We have performed BLASTn analysis between the two transcriptome and got hits around 700. Irrespective of changing parameters we could get hits between 600 to 700. In order cross verify we have performed alignment of (Raw reads from the database) transcriptome of Somvanshi et al., 2016 against the draft genome published by Bhat et al., 2022.

We have also mapped the transcriptome of Somvanshi et al., 2016 over the draft genome of Bhat et al., 2022. We have got similar results i.e. 1 % mapping.

The comparative analysis is included in the results section of the revised manuscript.

Response to comment 3: The genome submitted by Bhat et al. (2022) is a draft one without any annotations. We have mapped our transcriptome by using the draft genome as reference through GMAP bioinformatics tool. In reference-based assembly, 78,454 transcripts were mapped to the draft genome. Also, we have assembled our raw data from both the conditions with Draft genome of Bhat et al. (2022) as reference genome using HISAT2 bioinformatics tool. We have found that the 73.79% of reads from control sample assembled against the genome, while 90.29% of reads from anhydrobiotic nematodes sample assembled against it. These details are included in the revised manuscript.

Response to comment 4: As advised the Results and the Discussion have been modified

Response to comment 5: As advised the sections have been merged accordingly

Response to comment 6: The data has been submitted in NCBI database

BioProject ID: PRJNA789679

Bio sample No.: SAMN24146495

SRA IDs: SRR17284459, SRR17284460

---

## [Decision Letter · Decision Letter 1]

7 Jul 2022

PONE-D-21-38148R1Deciphering the mechanism of anhydrobiosis in the entomopathogenic nematode Heterorhabditis indica through comparative transcriptomicsPLOS ONE

Dear Dr. Mohan,

Unfortunately, the original reviewers did not agree to review the revised manuscript and I had to restore it to the newer reviewers who highlighted a few new changes. Please go through the comments and respond accordingly.  Please submit your revised manuscript by Aug 21 2022 11:59PM. If you will need more time than this to complete your revisions, please reply to this message or contact the journal office at plosone@plos.org. Please include the following items when submitting your revised manuscript:A rebuttal letter that responds to each point raised by the academic editor and reviewer(s). You should upload this letter as a separate file labeled 'Response to Reviewers'.A marked-up copy of your manuscript that highlights changes made to the original version. You should upload this as a separate file labeled 'Revised Manuscript with Track Changes'.An unmarked version of your revised paper without tracked changes. You should upload this as a separate file labeled 'Manuscript'.If applicable, we recommend that you deposit your laboratory protocols in protocols.io to enhance the reproducibility of your results. Protocols.io assigns your protocol its own identifier (DOI) so that it can be cited independently in the future. For instructions see: https://journals.plos.org/plosone/s/submission-guidelines#loc-laboratory-protocols. Additionally, PLOS ONE offers an option for publishing peer-reviewed Lab Protocol articles, which describe protocols hosted on protocols.io. Read more information on sharing protocols at https://plos.org/protocols?utm_medium=editorial-emailutm_source=authorlettersutm_campaign=protocols.

We look forward to receiving your revised manuscript.

Kind regards,

Pankaj Bhardwaj, Ph.D.

Academic Editor

PLOS ONE

Journal Requirements:

Reviewers' comments:

Reviewer's Responses to Questions

**Comments to the Author**

1. If the authors have adequately addressed your comments raised in a previous round of review and you feel that this manuscript is now acceptable for publication, you may indicate that here to bypass the “Comments to the Author” section, enter your conflict of interest statement in the “Confidential to Editor” section, and submit your "Accept" recommendation.

Reviewer #1: (No Response)

Reviewer #3: (No Response)

Reviewer #4: All comments have been addressed

2. Is the manuscript technically sound, and do the data support the conclusions?

Reviewer #1: Yes

Reviewer #3: Partly

Reviewer #4: Yes

3. Has the statistical analysis been performed appropriately and rigorously? 

Reviewer #1: Yes

Reviewer #3: No

Reviewer #4: Yes

4. Have the authors made all data underlying the findings in their manuscript fully available?

Reviewer #1: Yes

Reviewer #3: Yes

Reviewer #4: (No Response)

5. Is the manuscript presented in an intelligible fashion and written in standard English?

Reviewer #1: Yes

Reviewer #3: No

Reviewer #4: Yes

6. Review Comments to the Author

Reviewer #1: (No Response)

Reviewer #3: The manuscript titled "Deciphering the mechanism of anhydrobiosis in the entomopathogenic nematode Heterorhabditis indica through comparative transcriptomics" described the transcriptome comparison of anhydrobiotic and unstressed Heterorhabditis indica. For this purpose, the authors isolated total RNA for H. indica subjected to dehydration with a glycerine-water solution for 48 h. Then, a standard RNASeq pipeline was applied, and the data indiacted a upregulation of major, well known pathways linked to anhydrobiosis, including heat shock tolerance, polyamine biosynthesis, antioxidant defenses, among many others. Despite the importance of the subject of the manuscript, some major questions impair the manuscript publication in its current form.

Major points:

1. Despite the authors simulate a desiccation process by using a glycerine:water solution, the glycerol itself can be itself as a kosmotrope and chaotrope substance, which depends on the final concentration [Timson, David J. "The roles and applications of chaotropes and kosmotropes in industrial fermentation processes." World Journal of Microbiology and Biotechnology 36.6 (2020): 1-13]. In the experimental setup made by the authors, it is not clear if a ~10% glycerol solution could be inducing a dehydration response per se. Why not lowering the humidity of the chamber (below 60%) to induce a dehydration response?

2. The description of GOs and its graphical representations are below the standard (Figures 4 and 5). What are the statisticals related to GO analysis (number of mapped transcripts, p-value, FDR)? A table containg the GO data and all statistical analysis is preferable instead of pizza graphics.

3. The authors performed a clustering analysis of qPCR data (Figures 7 and 8)? If not, I strongly advise in order to better see the clusters of similar expressed genes.

Reviewer #4: (No Response)

7. PLOS authors have the option to publish the peer review history of their article (what does this mean?). If published, this will include your full peer review and any attached files.

Reviewer #1: No

Reviewer #3: No

Reviewer #4: No

---

## [Author Response · Author response to Decision Letter 1]

1 Sep 2022

Comment 1- In the experimental setup made by the authors, it is not clear if a ~10% glycerol solution could be inducing a dehydration response per se. Why not lowering the humidity of the chamber (below 60%) to induce a dehydration response?

Response: The preconditioning of juveniles in ~10% glycerol solution (97% RH), over a period of 2 days, helps in a slow and gradual adaptation towards anhydrobiosis. The juveniles enter into desiccated state and after which they can survive in the almost complete absence of water and revive upon rehydration. 

These nematodes are very sensitive to water loss and only gradual water loss, over a period of time, enable the morphological and physiological adjustments to help nematodes enter anhydrobiosis. At lower RH (below 90%) the water loss is rapid which does not allow nematode metabolism to regulate timely. Hence, nematode exposed to lower RH dry quickly and do not adapt to survive well.

“…During entry into anhydrobiosis, a gradual water loss occurs over time, as water content falls from 75-80% in active nematodes to 2-5% in anhydrobiotic forms (Demeure Freckman 1981). Survival is best if nematodes dry slowly; most species are killed if drying occurs too quickly (Barrett 1991; Demeure Freckman 1981)….”

Under natural field conditions, lack of rainfall and irrigation coupled with extended period of heat pulls the moisture away from the soil. The surface-soil starts to dry first, and the dryness extends gradually vertically into the A, B and O horizons. This allows Heterorhabditis and other anhydrobiotic nematodes, mostly prevalent in these horizons, enter into anhydrobiosis. 

Erkut (2011) induced similar anhydrobiosis in C. elegans at 98% RH over an extended period of 4 days using NaOH (Erkut, C., Penkov, S., Khesbak, H., Vorkel, D., Verbavatz, J. M., Fahmy, K., Kurzchalia, T. V. (2011). Trehalose renders the dauer larva of Caenorhabditis elegans resistant to extreme desiccation. Current biology: CB, 21(15), 1331–1336. https://doi.org/10.1016/j.cub.2011.06.064) 

Both, C. elegans and Heterorhabditis are closely related nematodes belonging to Order Rhabditida. Our lab is successfully using the protocol for inducing anhydrobiosis at 97% RH in Heterorhabditis juveniles (Manimaran et al., 2012, 2015). The reference has been added in the manuscript.

Comment 2- The description of GOs and its graphical representations are below the standard (Figures 4 and 5). What are the statisticals related to GO analysis (number of mapped transcripts, p-value, FDR)? A table containg the GO data and all statistical analysis is preferable instead of pizza graphics

Response: As per the referee’s comment the modifications in the analysis have been incorporated and representation has been made in a tabulated form (Figure 6, Supplementary Table 9)

Comment 3- The authors performed a clustering analysis of qPCR data (Figures 7 and 8)? If not, I strongly advise in order to better see the clusters of similar expressed genes.

Response: The clustering analysis has been already done

---

## [Decision Letter · Decision Letter 2]

14 Sep 2022

Deciphering the mechanism of anhydrobiosis in the entomopathogenic nematode Heterorhabditis indica through comparative transcriptomics

PONE-D-21-38148R2

Dear Dr. Mohan,

We’re pleased to inform you that your manuscript has been judged scientifically suitable for publication and will be formally accepted for publication once it meets all outstanding technical requirements.

Kind regards,

Pankaj Bhardwaj, Ph.D.

Academic Editor

PLOS ONE

Additional Editor Comments (optional):

Reviewers' comments:

Reviewer's Responses to Questions

**Comments to the Author**

1. If the authors have adequately addressed your comments raised in a previous round of review and you feel that this manuscript is now acceptable for publication, you may indicate that here to bypass the “Comments to the Author” section, enter your conflict of interest statement in the “Confidential to Editor” section, and submit your "Accept" recommendation.

Reviewer #3: All comments have been addressed

2. Is the manuscript technically sound, and do the data support the conclusions?

Reviewer #3: Yes

3. Has the statistical analysis been performed appropriately and rigorously? 

Reviewer #3: Yes

4. Have the authors made all data underlying the findings in their manuscript fully available?

Reviewer #3: Yes

5. Is the manuscript presented in an intelligible fashion and written in standard English?

Reviewer #3: Yes

6. Review Comments to the Author

Reviewer #3: The revised manuscript can be accepted for publication. All questions were addressed by the authors.

7. PLOS authors have the option to publish the peer review history of their article (what does this mean?). If published, this will include your full peer review and any attached files.

Reviewer #3: No

---

## [Editor Report · Acceptance letter]

6 Oct 2022

PONE-D-21-38148R2 

Deciphering the mechanism of anhydrobiosis in the entomopathogenic nematode *Heterorhabditis indica* through comparative transcriptomics 

Dear Dr. Mohan:

I'm pleased to inform you that your manuscript has been deemed suitable for publication in PLOS ONE. Congratulations! Your manuscript is now with our production department. 

Kind regards, 

on behalf of

Dr. Pankaj Bhardwaj 

Academic Editor

PLOS ONE